# Multi-scale Interaction Mechanism for Edge-Localized-Mode Suppression in the Tokamak Edge

Zeyu Li [1] ✉, P. H. Diamond [2] ✉, Xi Chen [1], F. Khabanov[3], Xueqiao Xu [4], R. J. Hong[5], V. S. Chan[6], C. M. Muscatello [1], L. Zeng[5], G. Y. Yu[7], T. Rhodes[5], G. R. McKee[3], Zheng Yan[3] & M. E. Austin [8]

A central challenge in fusion energy is reconciling the high-confinement mode required for reactor performance with the intense intermittent relaxation events it produces, known as edge-localized modes. These instabilities arise in the steep pressure pedestal at the plasma edge when magnetohydrodynamic thresholds are crossed, inflicting damaging heat loads on reactor components. Here, we show that multiscale interactions between microscopic turbulence and macroscopic magnetohydrodynamic modes provide encouraging prospects for self-organized edge-localized modes regulation. Using direct quantitative measurements of multiscale modes, eddy dynamics, and turbulent flux, we show that small-scale electron drift wave turbulence actively scatters the large-scale peeling-ballooning modes. This scattering decorrelates the pressure and velocity fields of the instability, so arresting its growth. Our modeling and theoretical analysis confirm this suppression mechanism is effective even when conventional linear stability thresholds are exceeded. This work establishes a nonlinear principle for edge-localized modes stability, revealing how ambient micro-turbulence can be leveraged to maintain a macro-stable, high-performance pedestal for future fusion reactors.

The study of bursty, explosive relaxation events is a captivating subject that spans various areas of physics, from solar flares and magnetic substorms in space physics contexts[1,2] to edge-localized modes (ELMs) in magnetically confined fusion plasmas[3]. They turn out to share a common qualitative picture under very different physical conditions. Often, relaxation events can be explained by identifying the exceedance of the linear stability threshold. Conversely and yet unresolved, is how a high confinement system can exist without relaxation events, even when its stored energy exceeds the known stability threshold. The edge of a fusion tokamak plasma has shown both relaxation events and quiescent behavior under similar background conditions. Focusing our study on this system, we hope to elucidate the underlying physics of the quiescent operation, with the possibility of generalizing it to other systems.

Research on relaxation events in the tokamak edge has practical motivation. ELMs, in particular, are a critical concern for the development of tokamak-based fusion power plants, one of the leading candidates for future clean energy sources. These violent, periodic events, occurring at the plasma edge in high-confinement regimes, can expel significant amounts of energy and particles, potentially causing severe damage to the plasma-facing components of the reactor[3]. Therefore, understanding and controlling ELMs is of paramount importance for the safe and sustained operation of future fusion reactors.

[1]General Atomics, San Diego, CA, USA. [2]University of California, San Diego, CA, USA. [3]University of Wisconsin-Madison, Madison, WI, USA. [4]Lawrence Livermore National Laboratory, Livermore, CA, USA. [5]University of California, Los Angeles, CA, USA. [6]Institute of Plasma Physics, Chinese Academy of Sciences, Hefei, Anhui, China. [7]University of California, Davis, CA, USA. [8]University of Texas at Austin, Austin, TX, USA. ✉e-mail: lizeyu@fusion.gat.com; pdiamond@ucsd.edu

Over the years, researchers have developed several active and passive methods to mitigate or suppress ELMs. Notable examples include the application of resonant magnetic perturbations (RMPs)[4] and the achievement of the Quiescent H-mode (QH-mode)[5–8]. The former involves the use of externally applied magnetic fields to reduce the edge plasma pressure, leading to the suppression of ELMs. The latter, on the other hand, aims to achieve ELM-free operation by maintaining a mildly turbulent plasma edge. A particular recent advance in the latter approach was the discovery of the "wide-pedestal" QH-mode on the DIII-D tokamak, which achieved ELM-free high-confinement operation with net-zero input torque, peeling-limited pedestal and other reactor-relevant parameters[9–12]. This unique regime exhibits a significantly broader pedestal width than predicted by conventional scaling laws. A key feature of the wide-pedestal QH mode is the presence of both broadband MHD activity and intermittent turbulence within the pedestal, which are observed to rotate in the ion diamagnetic drift direction and the electron diamagnetic drift direction, respectively[11]. Supporting the experimental observation, reduced two-fluid modeling using the BOUT++ code has also identified two key components: a large-scale peeling-ballooning mode (PBM) and a small-scale electron drift wave (EDW)[13]. Importantly, the characteristics of these fluctuations are intimately linked to the structural features of the pedestal itself, suggesting a complex interplay between the pedestal structure and the stability of the plasma edge[11,14]. The study of wide-pedestal QH mode offers insights into suppressing relaxation events in general, and a promising operation regime for reducing transient heat load to the wall in future tokamaks[12,15].

To date, a complete understanding of the underlying physics that governs ELM stability and the transition to quiescent regimes remains an active area of research. Prior numerical studies have investigated the impact of multi-scale turbulence on core confinement using gyrokinetic simulations[16–19], and the interaction between magnetic islands and microturbulence has been widely explored[20–24]. At the edge, several simulations have examined the evolution of PBMs during ELM saturation[25], while others have studied nonlinear coupling between disparate modes via poloidal flows in QH mode[26–28]. Experimentally, signs of multi-scale interaction and pedestal regulation have been reported across different devices and scenarios[29–34], including suggestions of three-wave coupling[29]. However, a clear picture of how small-scale turbulence interacts with large-scale MHD modes, especially its role in modifying the MHD transport and mediating ELM dynamics, remains lacking. In particular, the precise mechanisms that sustain wide-pedestal QH-mode plasmas in a quiescent state continue to be a subject of active investigation.

In this paper, we present, for the first time, a comprehensive analysis of the interplay between scale-separated modes and their impact on ELMs, leveraging a synergistic combination of experimental observations, numerical simulations, and theoretical modeling. Advanced diagnostics on the DIII-D tokamak, including Beam Emission Spectroscopy (BES) velocimetry[35], allow direct observation of the interactions between large-scale PBMs and small-scale EDWs. Further analysis of the turbulent transport flux provides crucial insights into the impact of these interactions on particle transport. We find that the cross-phase between density ($\tilde{n}$) and radial velocity ($\tilde{v}_r$) fluctuations of the large-scale MHD mode is scattered during bursts of EDW turbulence. This scattering significantly reduces the outward turbulent particle flux. Supporting these experimental findings, BOUT++ simulations[36] indicate that the small-scale EDWs effectively scatter the cross-phase[37] between the pressure ($\tilde{p}$) and radial velocity ($\tilde{v}_r$) perturbations of the PBM, leading to the suppression of PBM transport and relaxation. By incorporating the EDW phase scattering effect into our theoretical analysis, we determine that the nonlinear ELM onset boundary is upshifted compared to predictions from linear peeling-ballooning theory. This nonlinear upshift in the peeling-ballooning stability boundary underpins a robust mechanism for maintaining a

turbulent yet quiescent pedestal, free of ELMs. This study thus elucidates the complex interplay between experimentally observed scale-separated modes and their effects on edge stability and ELM dynamics, offering insights that are pertinent not only to a broad spectrum of edge scenarios in fusion plasmas but also to other meta-stable physical systems governed by multi-scale interactions.

## Results
### Experimental observations and evidence of multi-scale MHD-turbulence interactions
The interplay of the scale-separated modes, each rotating in different poloidal directions in the plasma frame, is a prevalent phenomenon in the DIII-D wide-pedestal QH mode. The dataset presented in Fig. 1 is from a DIII-D wide-pedestal QH mode shot, featuring an isolated ELM at 2373 ms. The scale-separated modes appear in diverse diagnostics, with a low-frequency, $f = 10–50$ kHz, $k_\theta < 0.3$ cm$^{-1}$, MHD mode observed in BES; and a high-frequency, $f = 0.5–2.0$ MHz, $k_\theta = 2$-$4$ cm$^{-1}$, turbulence observed in DBS. In Fig. 1a, we observe the alternating increments in the frequency-integrated amplitudes of the MHD mode and turbulence. The large-scale ion-directed mode amplitude peaks between the adjacent peaks of the small-scale electron-directed mode. This results in an out-of-phase relationship, quantified by a Pearson correlation coefficient of $-0.33$. Both the pedestal electron density gradient ($\nabla n_e$, Fig. 1b) and temperature gradient ($\nabla T_e$, Fig. 1c) exhibit periodicities matching those of the observed MHD and turbulence modes, consistent with the findings shown in Fig. 6 of ref. 38. Similar turbulence-MHD interactions are documented across numerous wide-pedestal QH-mode discharges (see Supplementary Fig. 1). The recurring pattern of interactions between scale-separated modes, accompanied by corresponding changes in pedestal profiles, strongly indicates their dynamic coupling and combined impact on pedestal transport.

Direct evidence of the interaction between the scale-separated MHD and turbulence has been observed. With the aim of examining the interplay between the small-scale turbulence and the large-scale MHD mode, we undertake an in-depth analysis centered on a singular turbulence mode burst. To ensure precision, we scrutinized a strong, clear turbulence mode within the wide-pedestal QH mode in Fig. 2a. Within this interval, we can observe two bands consisting of the long-lasting MHD and the intermittent turbulence. Notably, the high-frequency turbulence exhibits a frequency down-chirping. The turbulence dominates at ~4562 ms. Later on, the MHD mode starts to dominate, with a steeper pedestal gradient, while the electron turbulence becomes weak, associated with the increment of the local diamagnetic flow shearing rate[14]. A robust bicoherence, well-above-noise level, is observed in the turbulence dominant phase, between 4561 and 4563 ms (see Supplementary Fig. 2). This indicates a strongly nonlinear interaction between the MHD and turbulence. To gain deeper insights into the dynamics of how the turbulence interacts with the MHD mode, we leverage the BES velocimetry technique[35,39]. This technique enables the measurement of the perturbated radial velocity $\tilde{v}_r$ and the normalized turbulent particle flux $\langle (\tilde{n}/n)\tilde{v}_r \rangle$. The turbulent particle flux results are presented in Fig. 2b. During the MHD dominant phase, we observed radially outward particle transport with a positive turbulent particle flux. Conversely, when turbulence dominates, the turbulent flux approaches zero or even becomes negative, indicating that the outward transport driven by the low-frequency mode (10–60 kHz) is suppressed. These findings align with the observation of steepened electron density pedestal, as shown in Fig. 2d. We can see the electron pedestal profile steepens shortly after the turbulence mode bursts in the ensemble profiles (in Fig. 2c). These results are corroborated by an in-depth analysis of the 0.2 ms window of the raw turbulence data in Fig. 2e, f. In the MHD dominant phase, Fig. 2e, one can see that the normalized density perturbation $\delta n$ mostly exhibits an in-phase relation with the radial velocity perturbation $\delta v_r$, and so drives a positive (outward) flux across the pedestal. Meanwhile, in the turbulence

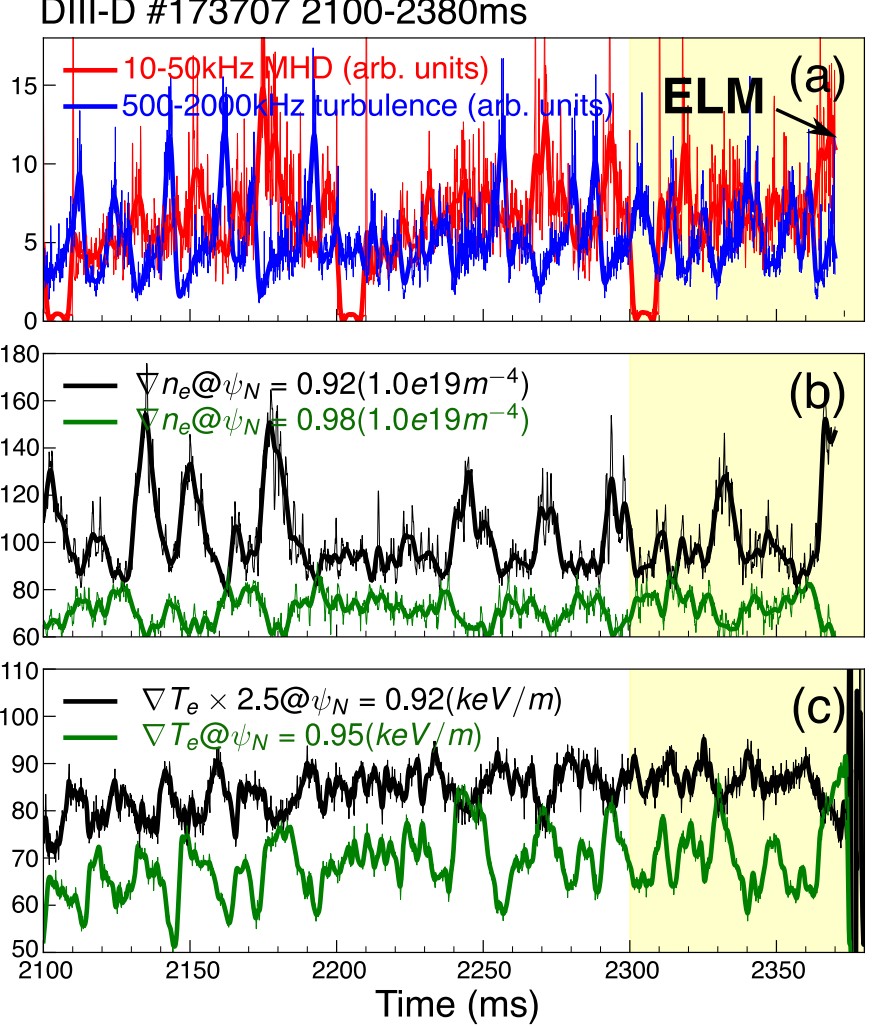

**Fig. 1 | Temporal evolution of edge turbulence and pedestal gradients in DIII-D wide-pedestal QH mode discharge #173707. a** low-frequency MHD mode measured by Beam-Emission-Spectroscopy[51] (BES, red) in the pedestal $\psi_N \sim 0.95$ and high-frequency turbulence measured by Doppler-Back-Scattering[52] (DBS, blue) in the pedestal $\psi_N \sim 0.92$, where $\psi_N \sim 0.90$ is the pedestal top and $\psi_N \sim 1.0$ is the separatrix, as depicted in Fig. 2d; **b** electron density gradient at $\psi_N = 0.92$ (black) and 0.98 (green) measured by Reflectometry[53]; **c** electron temperature gradient at $\psi_N \sim 0.91$ (black) and 0.96 (green) measured by ECE[54]. The raw data is shown in faint color, and the solid lines are the low-pass filtered signals. The shaded region indicates the period approaching the ELM event.

dominant phase (Fig. 2f), the cross-phase between the $\delta n$ and $\delta v_r$ is strongly scattered and even shows an out-phase relation. This collectively contributes to a negative (inward) flux. These observations underscore the critical role of the cross-phase between perturbed density and radial velocity in regulating transport during multi-scale MHD-turbulence interactions.

To further illustrate the contrasting spatial structures and transport characteristics of the two phases, we map the two-dimensional density fluctuation and corresponding velocity vector field in the R-Z plane using BES velocimetry. In Fig. 3a1–i1, taken during the low-frequency, MHD-dominant interval, the eddies extend across the radial and poloidal extent of the cross-section, and rotate in the ion-diamagnetic drift (downward) direction. Notably, the positive perturbations (red) collectively shift outward and negative perturbations (blue) move inward, indicating strong outward radial transport. In contrast, in Fig. 3a2–i2, taken during the turbulence-dominated phase, reveals more compact structures with reduced radial and poloidal extent, rotating in the electron-diamagnetic drift (upward) direction. The associated velocity vectors indicate primarily poloidal (tangential) motion with complex radial displacement, unlike the MHD phase, positive eddies are not consistently moving outward, nor negative ones inward. These observations are consistent with negligible net

radial transport and scattered cross phase, as depicted in Fig. 2b. These observations provide direct experimental evidence that turbulence reduces eddy size, reverses the mode rotation direction, and, critically, suppresses the cross-phase term $\langle \delta n \delta v_r \rangle$, thereby nonlinearly inhibiting large-scale MHD-induced transport.

## Impact of Multi-Scale Interactions on ELM Dynamics and Modeling Support

The interplay between scale-separated modes plays a vital role in determining the edge states and ELM dynamics. In Fig. 4, an isolated ELM in wide-pedestal QH mode is observed, offering an opportunity to investigate the underlying mechanisms of multiscale mode interaction on ELM dynamics. The low-frequency MHD mode, evident in both magnetics and density fluctuations, Fig. 4b, c, exhibits an electromagnetic signature. One can see that the amplitude of the low-frequency MHD mode is reduced when the high-frequency turbulence becomes strong, i.e., at 2340 ms and 2360 ms. Meanwhile, as the ELM crash approaches, at 2363-2373 ms, the high-frequency turbulence weakens, and the low-frequency MHD mode becomes dominant, coincident with the development of the steepened edge density gradient (Fig. 1b and Supplementary Fig. 3). The ELM ultimately occurs when two conditions are satisfied: (a) a sufficiently steepened pedestal

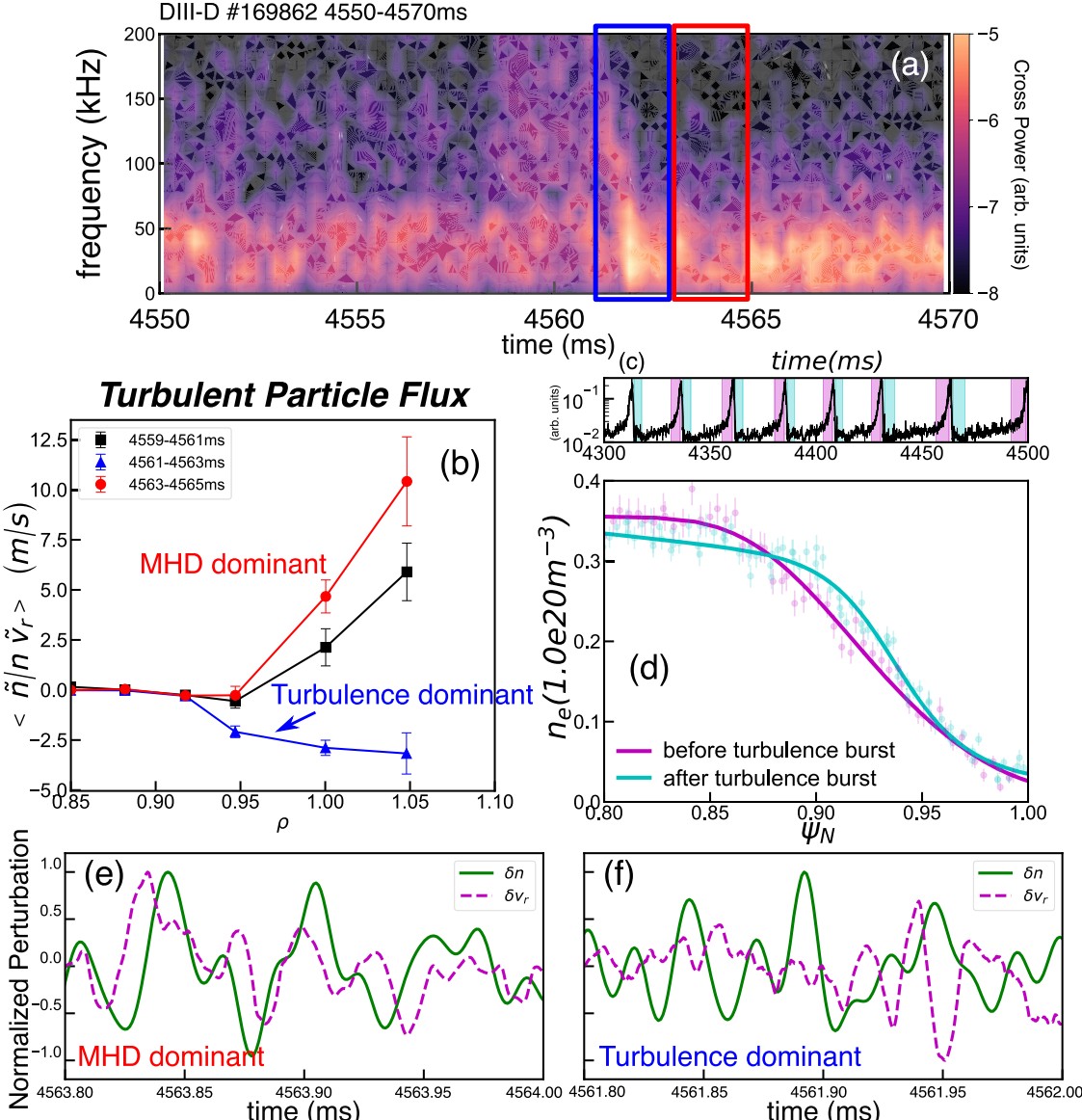

**Fig. 2 | The zoomed-in plot of one strong turbulence mode burst and turbulent particle flux in DIII-D wide-pedestal QH mode discharge #169862. a** The cross-power of density perturbation obtained from two poloidally adjacent channels at $\psi_N \sim 0.94$ of the BES diagnostic. **b** Poloidal averaged turbulent particle flux, $\langle (\tilde{n}/n)\tilde{v}_r \rangle$, computed by integrating in the frequency range of 10–60 kHz from the BES velocimetry technique. Error bars represent the standard deviation of the measured values over the specified time period. **c** Time trace of the 0.5–2 MHz electron turbulence amplitude and the window selected for profile reconstruction. **d** Ensemble-averaged electron density profile at the pedestal, relative to the turbulence burst phase, measured using Thomson scattering. Error bars indicate systematic uncertainties. **e, f** are the evolution of normalized density perturbation $\delta n$ and perturbed radial velocity $\delta v_r$, near the separatrix $\psi_N \sim 1.0$, during the MHD dominant phase and the turbulence dominant phase, respectively. Positive $\delta v_r$ indicates radially outward and negative means inward.

profile that enhances the edge bootstrap current and pushes the plasma beyond the linear PBM stability boundary, and (b) a lack of strong turbulence–MHD interaction, allowing the MHD mode to grow and initiate the collapse. This behavior supports the hypothesis that electron-scale turbulence can nonlinearly suppress low-n peeling–ballooning modes and their transport, effectively expanding the ELM-stable operating space.

Having established evidence that the turbulence-MHD interaction relates to the ELMs in wide pedestal QH mode, we now turn to understanding the physical mechanism. To this end, BOUT + + reduced two-fluid three-field[36] simulations are employed, which evolves the pressure perturbation $\tilde{p}$, vorticity $\tilde{U}$, and magnetic vector potential $\tilde{A}_{\parallel}$. BOUT + + three-field module can capture the essence of the interaction of scale-separated modes by incorporating the parallel electron dynamics into the generalized Ohm's law[13]. Within this framework,

BOUT + + identifies the large-scale electromagnetic MHD mode as PBM and the small-scale electron turbulence as the drift Alfvén wave. We label the latter here as an electron drift wave (EDW), without loss of generality.

Numerical modeling supports the hypothesis that small-scale electron drift waves influence the large-scale PB mode, thereby leading to various ELM states through the scattering of the cross-phase between pressure and radial velocity perturbations. BOUT + + linear simulation reveals that PBM dominates the low-n ($n = 3$–20) branch, while EDW dominates the high-n ($n > 20$) branch, as depicted in Fig. 5a. The radial mode structure shown in Fig. 5b, illustrates that the small-scale EDW originates in the upper pedestal during the linear phase and subsequently spreads radially inward and outward during nonlinear saturation, consistent with experimentally observed mode structures[11,13]. We investigate the interplay between PBM and EDW

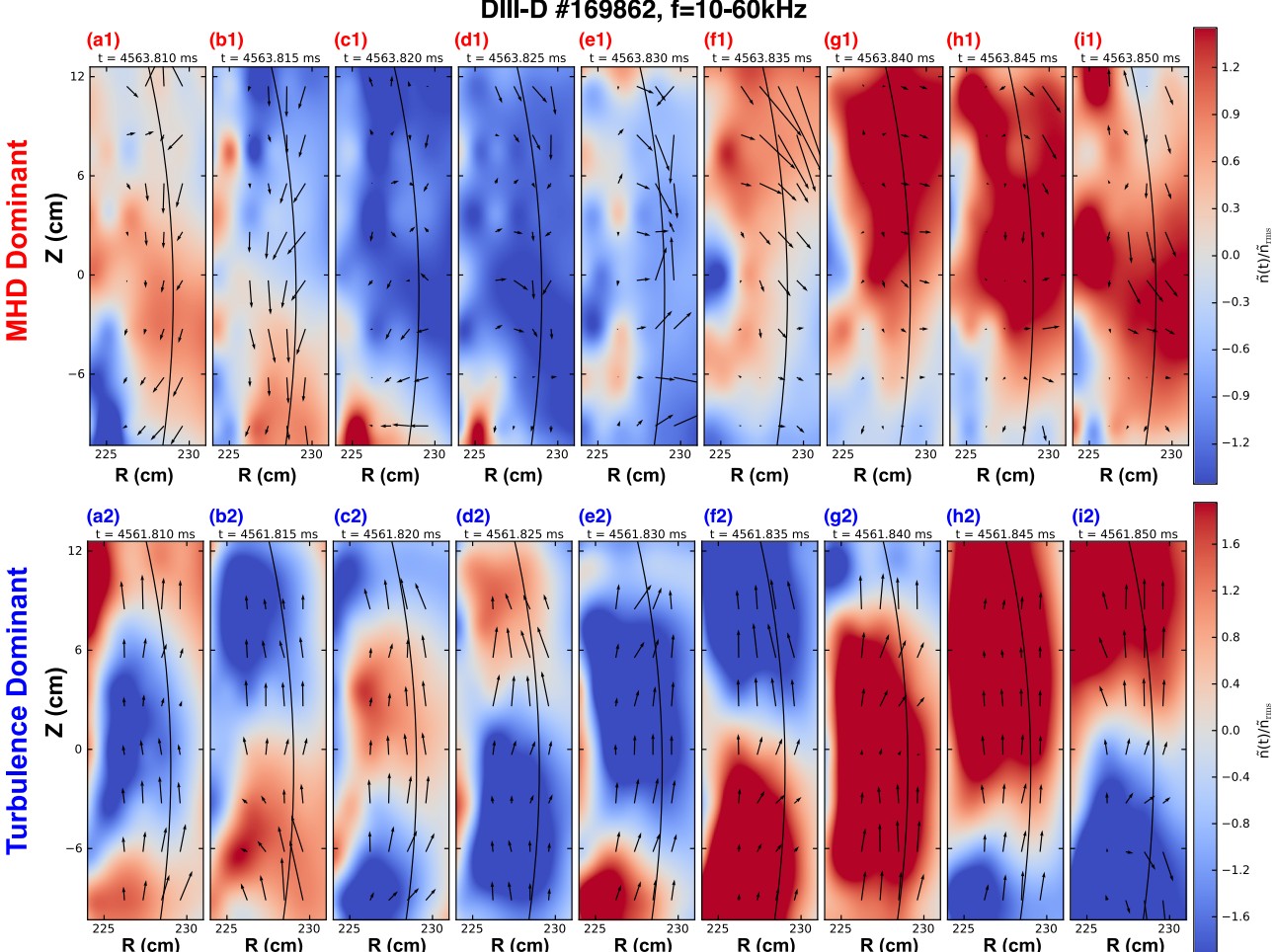

**Fig. 3 | Two-dimensional evolution of edge density fluctuations and velocity-field vectors in MHD- and turbulence-dominant phases of DIII-D discharge #169862 (10-60 kHz).** Top-row panels (**a1**–**i1**) show successive $5\mu$ s-interval snapshots of the normalized density fluctuation $\tilde{n}/\tilde{n}_{rms}$ in the R–Z plane at t = 4563.810–4563.850 ms, during a low-frequency, MHD-dominant interval. Bottom-row panels (**a2**–**i2**) show the same sequence at t = 4561.810–4561.850 ms, when electron-drift-wave turbulence dominates. In each frame the blue-to-red shading denotes negative-to-positive density perturbations, the black curve marks the normalized separatrix ($\psi_N = 1$), and overlaid arrows are normalized velocity vectors ($\tilde{v}_r, \tilde{v}_z$) averaged over $\pm 1\mu$ s around each time. The arrow length reflects the amplitude of the velocity field at each location. During the MHD dominant phase the eddies are broad, rotate in ion-diamagnetic (downward) direction and correlate with strong radial transport, whereas in the turbulence dominant phase they become poloidally and radially scattered, rotate in electron-diamagnetic drift (upward) direction and produce negligible radial flux.

through numerical experiments in BOUT + +. One case includes both PBM and EDW by retaining $n$ modes ranging from $n = 5,10,15,...,80$ in the BOUT + + nonlinear simulation, represented by the blue curve in Fig. 5c, d. In another case, we filtered out the high-$n(n > 20)$ EDW modes, to simulate the case with PBM alone. A notable difference arises during the final saturation stage, as shown in Fig. 5c. When both the PBM and EDW are retained, the high-$n$ pressure perturbations initially grow and saturate at a lower amplitude (Supplementary Fig. 4), subsequently interacting with the linearly most unstable PBM ($n = 10$). The $n = 10$ PBM saturates at $\delta p/p_0 \sim 0.06$, closely matching the experimental observation, and the pedestal remains quiescent. Conversely, in the case of retaining only PBM, the PBM continues unperturbed growth and eventually saturates at a much higher level ($\delta p/p_0 \sim 0.23$), leading to a pedestal crash and the occurrence of an ELM (as seen in Supplementary Fig. 4b, c). Figure 5d, e demonstrate that the cross-phase between $\delta p$ and $\delta v_r$ is strongly scattered in the presence of the EDW. The numerical findings on phase scattering due to EDW turbulence closely align with experimental observations shown in Fig. 2. Furthermore, the simulated mode eddy evolution (Supplementary Figs. 5 and 6) captures the essential behavior observed in Fig. 3, including shrinkage of edge structures, reversal of

rotation direction, and suppression of radial transport. The pedestal remains quiescent when turbulence-MHD interaction is included, whereas the PBM-only case results in strong transport. Taken together, these results provide compelling evidence that the turbulence suppresses the large-scale PBM. This aligns with the critical role of the cross-phase in determining the ELM state, as discussed in refs. 25 and 37, with a specific emphasis on the multi-scale mode interaction within this study. The interplay of scale-separated modes thus crucially influences pedestal stability, elaborating with implications for nonlinear ELM dynamics.

## Theoretical Understanding and Application to General ELM Physics

We establish a theoretical ELM onset boundary by considering the influence of small-scale electron drift wave scattering on the large-scale PBM. This model is motivated by experimental observations (Fig. 2) and numerical simulations, which confirm that electron-scale turbulence can grow and saturate in the presence of strong E×B flow and PBM activity. We develop a model to illustrate and quantify the impact of the electron drift wave scattering effect on peeling-ballooning modes. Utilizing the same set of the three-field equations,

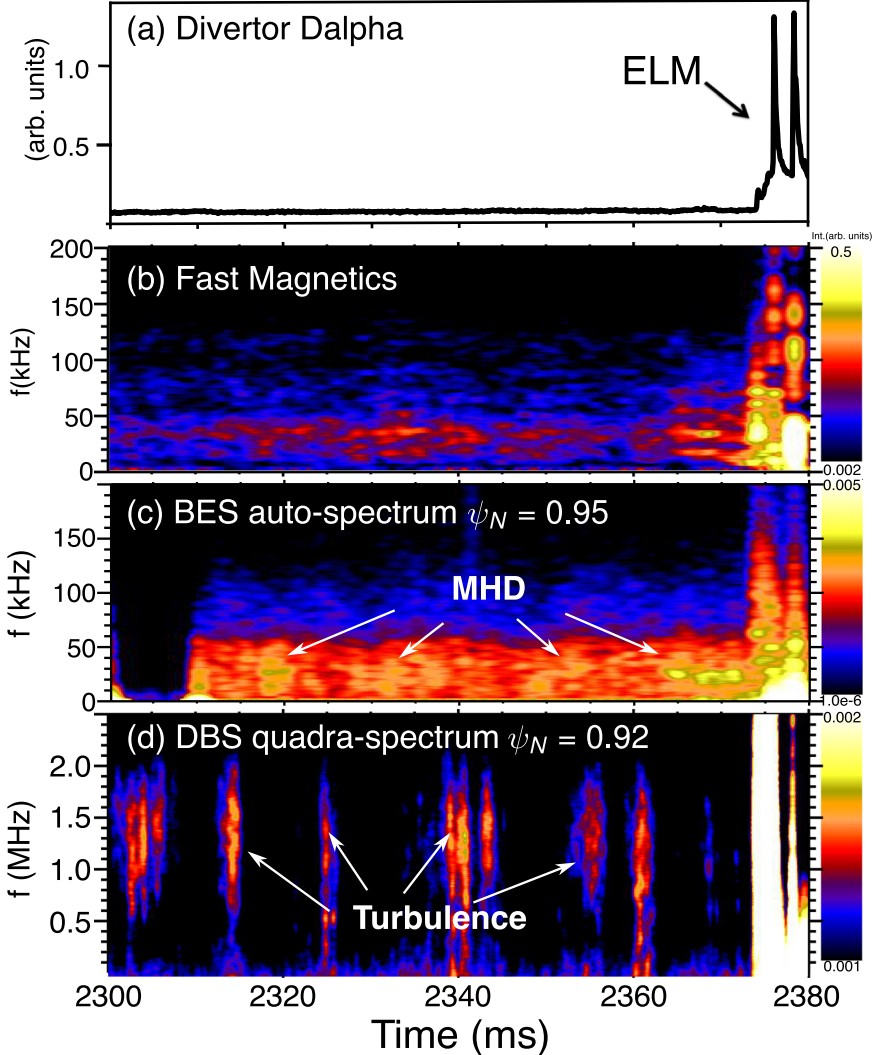

**Fig. 4 | ELM burst and associated MHD-turbulence activity in DIII-D wide-pedestal QH-mode discharge #173707. a** Divertor $D\alpha$ signal indicating the timing of the ELM crash; **b** Magnetic fluctuations measured by high-resolution magnetic probes; **c** Broadband (0–200 kHz) electron density fluctuations measured by beam emission spectroscopy (BES); **d** High-frequency (0–2.5 MHz) density fluctuations measured by Doppler backscattering (DBS). This is the same discharge as shown in Fig. 1.

we retained the nonlinear terms of the EDW to study the PBM mode growth within the presence of saturated EDW. By assuming the $\widetilde{p}_{PB} \ll \widetilde{p}_{DW}$ (i.e., the PBM is developing), the three field equations become:

$$\frac{\partial}{\partial t}\widetilde{p}_{PB} + \widetilde{v}_{DW} \cdot \nabla \widetilde{p}_{PB} = -\widetilde{v}_{PB} \cdot \nabla p_0 \tag{1}$$

$$\frac{\partial}{\partial t}\widetilde{U}_{PB} + \widetilde{v}_{DW} \cdot \nabla \widetilde{U}_{PB} = B_0 \nabla_\parallel J_\parallel + 2\vec{b}_0 \times \vec{\kappa}_0 \cdot \nabla \widetilde{p}_{PB} \tag{2}$$

$$\frac{\partial}{\partial t}\widetilde{A}_{\parallel,PB} + \widetilde{v}_{DW} \cdot \nabla \widetilde{A}_{\parallel,PB} = -\nabla_\parallel \widetilde{\phi}_{PB} + \frac{1}{2en_0}\nabla_\parallel \widetilde{p}_{PB} + \frac{\eta}{\mu_0}\nabla_\perp^2 \widetilde{A}_{\parallel,PB} \tag{3}$$

Here, the quantities are the same as the settings in ref. 36 where $J_\parallel = J_{\parallel 0} - \frac{1}{\mu_0}\nabla_\perp^2 \widetilde{A}_\parallel$ represents the parallel current, $n_0$ denotes equilibrium density, $\vec{\kappa}_0 = \vec{b}_0 \cdot \nabla \vec{b}_0$, and $\eta$ corresponds to the realist Spitzer-Härm resistivity. The subscripts of PB and DW denote quantities for the PBM or EDW, respectively. As seen, the right-hand side terms remain the same as for the prior linear results, meaning the PBM drive terms

are identical. Note, $\widetilde{v}_{DW}$ here defines the random velocity fluctuations of the ambient drift wave turbulence. These scatter the coherence of the peeling-ballooning flux. Through Fourier decomposition, we can obtain:

$$\frac{\partial}{\partial t}\widetilde{p}_k + N_k = -\widetilde{v}_k \cdot \nabla p_0 \tag{4}$$

where $N_k = [\widetilde{v} \cdot \nabla \widetilde{p}]_k = ik\sum_{k'}\widetilde{v}_{-k'}\widetilde{p}_{k+k'}$, and taking $\widetilde{p}_{k+k'} = \frac{-\widetilde{v}_{k'}ik\widetilde{p}_k}{-i(\omega+\omega')+\Delta\omega_{k+k'}}, \Delta\omega_{k+k'} \sim k'^2 D$ defines the decorrelation induced by ambient drift wave turbulence. Here $\omega'$, $k'_\perp$ are the drift wave frequency and wave vector, respectively. $\widetilde{D}_{DW}$ is the drift wave diffusion tensor and $D_{DW} = \int_0^\infty \langle \widetilde{v}_{DW}(0)\widetilde{v}_{DW}(\tau)\rangle d\tau$ following the Kubo formalism[40]. Then, we can get the relation between $N_k$ and $\widetilde{p}_k$, $N_k = d_k\widetilde{p}_k$, where $d_k \cong k^2\sum_{k'}\frac{\widetilde{v}_{-k'}\widetilde{v}_k}{-i(\omega+\omega')+k'^2 D}$. As the electron drift wave is comparatively high frequency, the real part of $d_k$ is approximated as:

$$d_{k,real} \cong k^2\sum_{k'}\frac{|\widetilde{v}_k|^2 k^2 D}{\omega'^2 + (k^2 D)^2} = k^2 D$$. Thus, the drift wave diffusivity scales

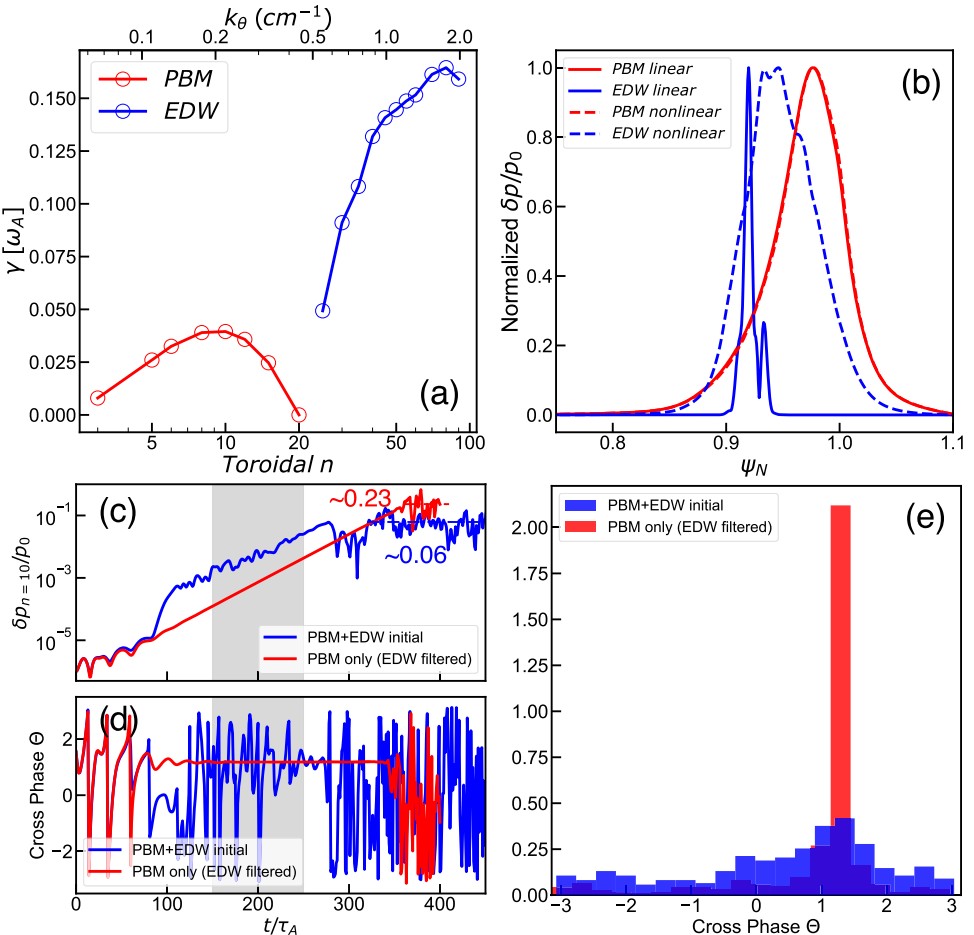

**Fig. 5 | BOUT++ numerical modeling of the wide-pedestal QH discharge #173707.** The edge bootstrap current is increased by 20%, compared with the stationary phase (2270–2370 ms), to simulate the dynamic approaching ELM crash. **a** Linear growth rate as a function of toroidal mode number ($n$); **b** Mode structure of PBM ($n = 10$) and EDW ($n = 50$) in the linear (solid) and EDW nonlinear saturation stage (dashed, gray-shaded region in Fig. 5c); Time traces of **c** the normalized pressure perturbation $\delta p/p_0$ and **d** the cross-phase between $\delta p$ and $\delta v_r$ for n = 10, with and without EDW, at $\psi_N = 0.95$, outer-mid-plane; **e** Probability density distribution of the cross-phase for cases with and without EDW, using the same time window $t = 0$–$400\tau_A$. The blue curves in these plots represent quiescent edge phases without ELM crashes, while the red curves correspond to periods with ELM crashes. The profile evolution, ELM associated energy loss, and mode eddy evolution are presented in Supplementary Fig. 4b, c, Supplementary Fig. 5 and Supplementary Fig. 6 for detailed reference.

as:

$$D \sim \frac{\left(\sum_{k'}\left(|\tilde{v}_{k'}|^2 k'^2\right) - \omega'^2\right)^{\frac{1}{2}}}{k'^2} \qquad (5)$$

Note that $D > 0$ requires a critical level of drift wave turbulence. This establishes that the onset of irreversible scattering of fluid elements by EDW requires the stochastic frequency of turbulent $E \times B$ advection to exceed the frequency of the EDW[41]. This observation aligns with the finding that the most pronounced interaction occurs when the drift wave chirps down to lower frequencies, as illustrated in Supplementary Fig. 2d. Substituting back into Eq. (4), we obtain a correction to the ideal peeling-ballooning growth rate $\gamma$, associated with the interaction with EDW:

$$\gamma + \triangle\gamma = \gamma + k_{PB}^2 D_{DW} = \gamma_{PB, ideal} \qquad (6)$$

Where $\triangle\gamma$ is the increment of the linear growth rate. There the diffusion tensor could be simplified as $D_{DW} \sim \frac{\langle v_{DW}^2 \rangle^{1/2}}{k}$. With Eq. (6), the level of scattering to shift the stability boundary corresponds to an effective increment in growth rate, $\triangle\gamma = k_{PB}^2 D_{DW}$. We can roughly estimate the

linear growth rate shift, as $\triangle\gamma = k_{PB}^2 D_{DW} \sim k_{PB}^2 \frac{\langle v_{DW}^2 \rangle^{1/2}}{k_{DW}}$, and $\langle v_{DW}^2 \rangle \sim \left(L_p \omega\right)^2 \left(\frac{\tilde{p}_{DW}}{p_0}\right)^2$. By assuming the drift Alfvén wave nature of the drift wave frequency, $\omega \sim k_\parallel V_{A0}$, one can get:

$$\frac{\triangle\gamma}{\omega_{A0}} \sim \left(\frac{k_{PB}}{k_{DW}}\right)(k_\parallel R)\left(k_{PB}L_p\right)\left\langle\left(\frac{\tilde{p}_{DW}}{p_0}\right)^2\right\rangle^{\frac{1}{2}} \qquad (7)$$

where $\omega_{A0}$ is the Alfvén frequency for normalization, $k_\parallel = 1/qR$ donates the parallel and perpendicular wave number; and $R$ and $L_p = [dln(p_0)/dr]^{-1}$ stand for the major radius and pressure scale length, respectively. Assuming the drift wave amplitude of $\tilde{p}_{DW}/p_0 \sim 5\%$, the estimated linear growth rate shift for n = 10 PBM, $\triangle\gamma/\omega_{A0} \sim 0.05$, which demonstrates reasonable agreement with BOUT++ numerical modeling. PBM experiences damping and suppression from the scattering effect, as depicted in the schematic plot Fig. 6. The pedestal pressure gradient and bootstrap current could grow beyond the linear growth rate boundary without triggering ELMs, thus defining an effective ELM onset boundary. The interaction between the EDW and PBM helps the turbulent pedestal of the wide-pedestal QH mode to remain quiescent.

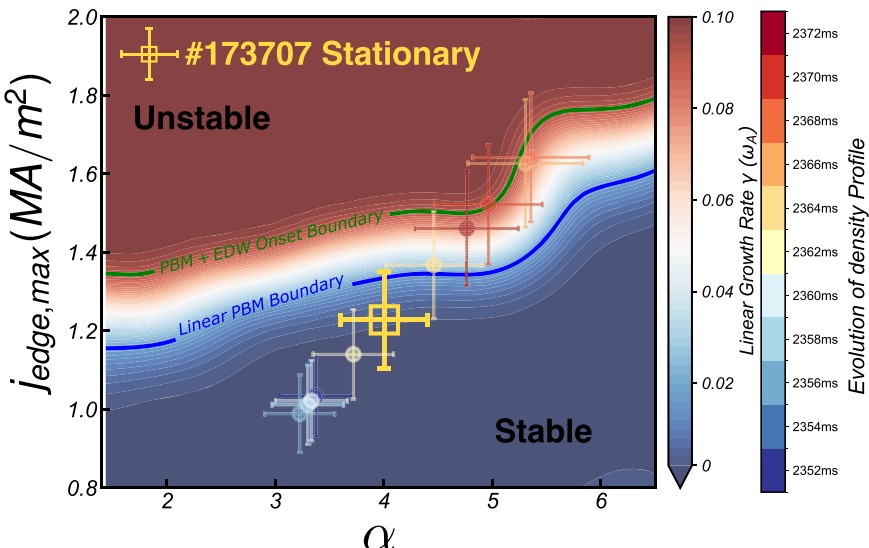

**Fig. 6 | Schematic of the peeling-ballooning stability boundary modified by small-scale drift wave turbulence scattering.** This figure illustrates how small-scale drift-wave scattering can expand the ELM-stable operational space. The $J - \alpha$ stability diagram is calculated for the stationary phase (2270–2370 ms) of QH-mode discharge #173707 using the VARYPED-ELITE suite[38,55], where $J$ is the peak edge current density and $\alpha = -2\mu_0 q^2 RP'/B^2$ is the normalized pedestal pressure gradient. The blue curve marks the standard linear PBM boundary, defined by a growth rate $\gamma = 0.02\omega_A$, while the green curve represents a hypothetical boundary where small-scale EDW scattering is assumed to weaken PBM drive without considering the self-consistent change in EDW intensity due to profile variation. The yellow square indicates the experimental operating point during the stationary wide-pedestal QH-mode phase. Colored symbols trace an illustrative trajectory approaching an ELM (2352–2373 ms), assuming $J$ and $\alpha$ scale with the measured electron density gradient evolution (see Supplementary Fig. 3). This trajectory is a schematic representation, not a precise calculation, due to the lack of high-temporal-resolution electron/ion temperature and ion density profiles. The right-most color bar shows the linear growth rate and the corresponding profile evolution time.

## Table 1 | Summary of the key findings of the multi-scale MHD-turbulence interaction in wide pedestal QH mode edge

| Physics Features | Experiment Evidence and Diagnostics | Theoretical and Modeling Insights |
|---|---|---|
| Multi-scale MHD and turbulence in tokamak edge | ☙ Large-scale, low-frequency: BES, magnetics, EM mode ($f$=10–50 kHz, $k_\theta < 0.3$ cm$^{-1}$)<br>☙ Small-scale, high-frequency turbulence: DBS, ($f$=0.5-2 MHz, $k_\theta = 2$–4 cm$^{-1}$) | ☙ Large-scale MHD: peeling-ballooning mode (PBM)<br>☙ Small-scale turbulence: electron drift wave (EDW) |
| Multi-scale mode interaction | ☙ Alternating amplitude growth between MHD and turbulence (Figs. 1a, 2a)<br>☙ Regulation of pedestal profiles (Figs. 1b, c, 2d)<br>☙ Strong, above-noise level bicoherence (Extended Fig. 2)<br>☙ Turbulence chirping down to lower frequencies impacts MHD-induced turbulent particle flux (Fig. 2b) | ☙ Large-scale PBM growth and saturation are influenced by small-scale EDW turbulence (BOUT++ modeling, Fig. 5c)<br>☙ Dependence of the required interaction amplitude on frequency shift in the diffusion tensor of theory model (Eq. (5)) |
| Interaction mechanism | ☙ Phase decoherence of $\delta n$ and $\delta v_r$ of MHD and the turbulent flux is reduced (BES velocimetry analysis, Figs. 2(e), 2(f))<br>☙ Shrinkage of edge structures, reversal of rotation direction, and suppression of radial transport (Fig. 3) | ☙ Cross-phase scattering of $\delta P$ and $\delta v_r$ observed in nonlinear BOUT++ modeling (Fig. 5(e), Supplementary Fig. 5, 6)<br>☙ Theoretical analysis reveals scattering of MHD by ambient drift wave turbulence (Eqs. (1)–(7)) |
| Mode interaction on ELM dynamics | ☙ As ELM onset approaches, large-scale MHD modes grow without suppression by small-scale turbulence (Figs. 4(a)–4(d)) | ☙ Significant differences in the saturation levels of perturbations with and without ambient EDW (Fig. 5(c), Supplementary Fig. 4)<br>☙ Drift wave scattering causes a expansion of PBM stable region (Fig. 6) |

## Discussion

In conclusion, our investigation highlights the interplay of scale-separated MHD-turbulences and how they regulate ELMs in DIII-D wide pedestal QH mode. We provided direct experimental evidence of multi-scale interaction through multiscale modes, eddy dynamics, and turbulent flux via advanced diagnostic capability. Specifically, we quantitatively established that the high-frequency, small-scale electron drift wave scatters the cross-phase between $\delta n/\delta P$ and $\delta v_r$ of the low-frequency, large-scale peeling-ballooning mode, thus causing decoherence of the cross-phase, and so mitigating the ELMs. Modeling and analysis support the concept of electron drift wave scattering induced decoherence of the turbulent particle and energy flux. Such consideration determines an ELM onset boundary that extends beyond the linear peeling-ballooning theory. The key physics findings are summarized in Table 1. It is noteworthy that, the density profile steepening, as depicted in Fig. 2d, may also overcome the ELM onset criterion with excessive electron drift wave strength, thus triggering ELMs. This observation raises fundamental questions about the mechanisms underlying density steepening, particularly whether electron-scale turbulence contributes to outward transport from the pedestal top and the generation of zonal flows, in addition to its role in suppressing MHD activity. These possibilities warrant further investigation in future studies.

Our findings highlight the critical role of a turbulent pedestal in extending the stable domain of peeling-ballooning modes. This study proposes the underlying physics of quiescent operation through the scattering of the multi-scale MHD-turbulence system, with potential applicability to other physical systems. Building on the fundamental mechanisms demonstrated here, along with recent research on turbulence spreading from the pedestal and its contribution to

broadening the divertor heat load width[42–45], the turbulent pedestal holds significant promise for application to ITER and future fusion reactors. These attributes underscore the pivotal role of pedestal turbulence in both ELM suppression and heat load broadening, positioning the turbulent QH mode as a particularly promising candidate for next-generation reactor designs.

## Methods

### DIII-D tokamak

The DIII-D tokamak is the largest magnetic fusion experimental device in the United States, supported by the U.S. Department of Energy Office of Science. The tokamak consists of a toroidal vacuum chamber surrounded by coils that produce the magnetic field to confine and shape the plasma. It has a major radius of 1.67 m and a minor radius of 0.67 m, with a toroidal magnetic field of up to 2.2 T. The plasma is created by applying a voltage to ionize a small amount of gas injected into the vacuum chamber and drive a large, toroidal electrical current. The plasma is then quickly heated to a high temperature by injection of high-power neutral beams (<16 MW), while additional gas fueling increases the density. More information can be found in ref. 46.

### BES measurements

The beam emission spectroscopy (BES)[35] diagnostic is applied for the study of dynamics of long-wavelength ($k_\perp < 3 cm^{-1}$) density turbulence structures on DIII-D. A 64-channel BES system is configured with an $8 \times 8$ grid of discrete channels that image an approximately $7 \times$ cm region at the outboard midplane with the sampling frequency of 1 MHz, thus providing a modest spatial resolution, high throughput, high-time resolution turbulence imaging system.

Turbulence velocity fields are obtained from BES images using velocimetry analysis based on an orthogonal dynamic programming (ODP) algorithm[35,47,48]. The velocimetry processes subsequent images of BES intensity fluctuations searching for local displacements and provides quantitative measurement of a two-dimensional flow-field in the lab frame. A velocimetry ODP algorithm is used to retrieve the velocity field from a sequence of BES turbulence images[35,48]. The 8 radial x 7 poloidal resolution of raw images is too low to be directly used by the ODP algorithm, so the images are spatially interpolated to a higher resolution of 40×40 using a radial basis function interpolation with a cubic norm. Velocity fields are calculated for 2 ms time slices and then averaged over the poloidal direction to get radial profiles. The innermost and the outermost points were excluded from the analysis due to larger errors in the output of ODP algorithm which can occur at the boundary of the image.

### BOUT + + Numerical modeling

BOUT + + reduced two-fluid three field module is employed for the numerical modeling of the wide-pedestal QH discharge #173707, which evolves the pressure perturbation $\tilde{p}$, vorticity $\tilde{U}$, and magnetic vector potential $\tilde{A}_\parallel$ [36,49] The simulation domain spans the pedestal and SOL regions, covering the normalized poloidal flux $\psi_N$ = 0.70-1.10, the resolution is set to $N_x = 512$, $N_y = 64$, and $N_z = 64$ in the radial, along the field line, and binormal direction on simulating a 1/5 of the torus. The simulation contains n = 5, 10, 15, …, 80 toroidal number modes. Realistic Spitzer-Härm resistivity is used in the modeling. The pedestal plasma profiles are based on the well-converged reconstructed equilibrium using EFIT code. The kinetic plasma profiles are measured between 2270-2370 ms for discharge #173707. To simulate the dynamic approaching ELM crash at 2373 ms, the edge bootstrap current is increased by 20% compared to the stationary phase, which is marginally unstable to the PB mode, as shown in Fig. 5(a). A realistic background radial electric field $E_r$ profile from the pedestal up to SOL, is calculated from the ion momentum balance equation based on the Charge Exchange Recombination (CER) Spectroscopy System in DIII-D and is used in the simulation.

## Data availability

The raw experimental data from the DIII-D National Facility that support the findings of this study are available from the corresponding author upon request. The source data and Python scripts required to reproduce the figures in this paper have been deposited in the Figshare repository[50].

## Code availability

The computer code used to generate results that are reported in the paper is available from the authors upon request.

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

## Acknowledgements

We thank the DIII-D Team for providing experimental data and operational support. Z.L. would like to thank Dr. A. Bortolon for revising and commenting on the manuscript, and Dr. N. Li, Dr. J. Dominguez-Palacios and Dr. X. Qin for useful discussions. This work was supported by the U.S. Department of Energy, Office of Science, Office of Fusion Energy Sciences, using the DIII-D National Fusion Facility, a DOE Office of Science user facility, under Award(s) DE-FC02-04ER54698 (all authors). This work was also supported by DE-FG02-04ER54738 (P.D.), DE-AC52-07NA27344 (X.Q.X.), DE-SC0019352 (R.H., T.R., L.Z.), DE-FG02-08ER54999 (F.K., G.M., Z.Y), DE-FG02-97ER54415 (M.A.), DE-FG02-99ER54531 (G.Y.) and LLNL-led SciDAC ABOUND Project SCW1832 (Z.L., P.D., X.Q.X.). This research used resources of the National Energy Research Scientific Computing Center, a DOE Office of Science User Facility supported by the Office of Science of the U.S. Department of Energy under Contract No. DE-AC02-05CH11231 using NERSC award FES-ERCAP0026742 (Z. L., X.Q.X.). P.D., Z.L., and X.C. would like to thank the Isaac Newton Institute for Mathematical Sciences, Cambridge, for support and hospitality during the Program, "Anti-diffusive dynamics: from sub-cellular to astrophysical scales", where some of the work on this paper was performed. This work is supported by an EPSRC grant, EP/RO14604/1.

## Author contributions

Z.L. led the experimental data analysis, numerical simulations, and manuscript writing. P.D. conceived the critical physics picture, guided the theoretical model development, and contributed to the manuscript writing. X.C. led the experimental demonstration and facilitated detailed discussions. X.Q.X. supported the BOUT + + code and guided the simulations. V.C. provided general discussions and assistance in organizing the manuscript. F.K., G.M., and Z.Y. contributed to BES data analysis for turbulence and velocimetry. R.H. and T.R. performed data analysis for DBS, while L.Z. conducted the analysis for Reflectometry.

M.A. provided guidance on using ECE data. C.M. and G.Y. offered general comments on the research.

## Competing interests

The authors declare no competing interests. This report was prepared as an account of work sponsored by an agency of the United States Government. Neither the United States Government nor any agency thereof, nor any of their employees, makes any warranty, express or implied, or assumes any legal liability or responsibility for the accuracy, completeness, or usefulness of any information, apparatus, product, or process disclosed, or represents that its use would not infringe privately owned rights. Reference herein to any specific commercial product, process, or service by trade name, trademark, manufacturer, or otherwise does not necessarily constitute or imply its endorsement, recommendation, or favoring by the United States Government or any agency thereof. The views and opinions of authors expressed herein do not necessarily state or reflect those of the United States Government or any agency thereof.
