## [Transparent Peer Review file · Nature Communications]

Multi-scale Interaction Mechanism for Edge-Localized-Mode Suppression in the Tokamak Edge

Corresponding Author: Dr Zeyu Li

Version 0:

Reviewer comments:

Reviewer #1

(Remarks to the Author)

The authors have answered all my questions. Thank you for your efforts! After completing the minor revision below, the paper is recommended for publication by me, without needing to see the revised version. Congratulations to the authors!

It is suggested to add a color bar corresponding to "The colored symbols" in Fig. 6, as in Extended Data Fig. 3 (a).

Reviewer #2

(Remarks to the Author)

Referee report on Ref: NCOMMS-25-49941-T "Multi-scale Interaction Mechanism for Edge-Localized-1 Mode Suppression in the Tokamak Edge"

General remark:

This manuscript presents interesting experimental observations and theoretical/numerical interpretations regarding multiscale interactions between magneto-hydrodynamic (MHD) and turbulence. However, several critical issues need to be addressed before it can be considered for publication.

1. General Critique on Novelty: The manuscript presents experimental results and analyses on wide pedestal quiescent high confinement mode in the tokamak edge, but they closely align with prior studies, offering limited divergence. Essentially, it reiterates previous observations without introducing significant novel insights or perspectives. This level of novelty is more suitable for a journal specialized in nuclear fusion than a multidisciplinary journal.

2. Novelty in MHD-Turbulence Interaction: The manuscript examines the nonlinear interactions between MHD and turbulence, which are expected to alter their respective properties. In MHD, these interactions typically modify cross-phase correlations, impacting transport. In turbulence, large-scale MHD-driven eddies are anticipated to suppress modes. However, these phenomena are well-documented in existing theoretical studies, and the manuscript offers no significant new contributions to this field.

3. Holistic Understanding of the Interaction Mechanism: A key aspect of MHD-turbulence interaction is understanding their holistic interplay and mutual influence. However, the manuscript does not deliver novel findings in this area. The experimental observations, as shown in Figures 1 and 3, merely document interaction patterns without providing new insights into the underlying dynamics. Moreover, prior studies, notably [14], have already thoroughly explored the fundamental aspects of this topic.

4. Issues with BOUT++ Simulations: The manuscript introduces BOUT++ simulations as a novel contribution, but these simulations suffer from significant shortcomings. The presented results show the growth of MHD-turbulence from arbitrarily imposed initial conditions. However, for the simulations to meaningfully contribute to experimental understanding, they must address the following:

o (i) Under what conditions pre-existing MHD structures allow turbulence to develop.

o (ii) How the developed turbulence subsequently modifies the properties of the existing MHD structures.

The omission of fundamental elements, such as the pedestal equilibrium ExB flow, renders these simulations little more than

a toy model with limited relevance to real experimental scenarios. In the referee's opinion, for this manuscript to meet the minimum novelty requirements of the journal to which it was submitted, it should at least provide answers to these issues.

5. Practical Relevance: The theoretical model presented in the final section also has limited applicability to experimental conditions. The model assumes a background turbulence field but does not explain how micro-turbulence with such characteristics can develop in the presence of macro-scale MHD structures. Specifically, it fails to account for how micro-turbulence could grow significantly enough to impact MHD dynamics within a pedestal region containing strong ExB flow and peeling-ballooning modes (or their precursor modes). Without addressing these critical aspects, the theoretical model lacks practical significance in the context of the experimental findings. This issue is critical to understand, as the authors claim, for applying the wide pedestal QH-mode as an operational mode for reactors.

Conclusion

Although the manuscript addresses an intriguing topic, its lack of novel experimental results, the inadequacy of simulations for meaningful physical interpretation, and the impracticality of the theoretical model raise significant concerns. Given these issues, the manuscript does not meet the publication standards of Nature Communications. The referee recommends rejection and suggests submitting to a specialized journal in thermonuclear fusion.

Version 1:

Reviewer comments:

Reviewer #2

(Remarks to the Author)

The authors have adequately addressed the referee's comments concerning the BOUT++ simulation, particularly regarding the E x B flow and the initial conditions.

I appreciate their efforts in performing additional simulations.

I believe that discussions and clarifications regarding the other parts of the manuscript have already been sufficiently carried out.

However, questions remain regarding the novelty of the work.

The idea of 'phase locking and phase slips in QH-mode' has been discussed in this field for more than a decade.

As the authors themselves repeatedly emphasize in their rebuttal, it is unclear whether this 'first direct measurement' truly provides new insights to the research community.

The referee acknowledges the authors' contribution in achieving the direct measurement and two-dimensional visualization. Nevertheless, I believe that this level of progress would be more suitable for a nuclear-fusion-specific journal, rather than a multidisciplinary one such as Nature Communications.

Therefore, the referee do not recommend publication in Nature Communications.

made.

Summary of Revisions

We thank the reviewers for their constructive and thoughtful feedback. We are pleased that Reviewer #1 found their concerns were fully addressed in the previous round and has recommended publication. We have implemented their minor suggestion to add a color bar to Fig. 6.

Regarding the comments from Reviewer #2, we note that they are **substantively the same as in the initial review round**. We believe our previous point-by-point rebuttal thoroughly addressed each of the concerns raised. In this submission, we have therefore taken the opportunity to build upon those original responses, further refining our arguments and providing additional details for clarity. We believe the revised manuscript now presents a clear and complete case for the mechanism and its significance.

To provide full context for the current state of the manuscript, we summarize these key improvements below:

1. We introduced a new figure (new Fig. 3) showing **unprecedented experimentally visual evidence** of how turbulence disrupts the edge MHD eddies.
2. We added **new experimental data** from multiple discharges across several years (new Extended Data Fig. 1) to demonstrate the universality of the observed mechanism;
3. We included a **new quantitative analysis** of the pedestal profile evolution leading to the ELM crash using high-resolution reflectometry data (new Fig. 6 and Ext. Data Fig. 3);
4. We have substantially clarified our theoretical and simulation sections, supported by **additional simulation results** (Ext. Data Figs. 4, 5, 6) that **reproduce key experimental dynamics**.

Below, we provide a point-by-point response to each comment. The referee's comments are shown in **bold black**, our responses in **purple**, and relevant manuscript text excerpts are marked in *blue italics*. The corresponding revisions are highlighted in the updated manuscript.

Response to Reviewer #1

The authors have answered all my questions. Thank you for your efforts! After completing the minor revision below, the paper is recommended for publication by me, without needing to see the revised version. Congratulations to the authors!

We sincerely thank Reviewer #1 for their highly positive and encouraging assessment. We are particularly grateful for their rigorous and insightful questions during the initial review round, which prompted us to make substantial improvements to the manuscript. We are very pleased that our comprehensive revisions successfully addressed all of the reviewer's initial concerns, leading to their strong recommendation for publication.

It is suggested to add a color bar corresponding to "The colored symbols" in Fig. 6, as in Extended Data Fig. 3 (a).

We thank the reviewer for this suggestion. We have added the color bar to Fig. 6 in the revised manuscript as requested.

Response to Reviewer #2:

General remark:

This manuscript presents interesting experimental observations and theoretical/numerical interpretations regarding multiscale interactions between magneto-hydrodynamic (MHD) and turbulence. However, several critical issues need to be addressed before it can be considered for publication.

We thank the reviewer for their time and for finding our manuscript of interest. While we appreciate the critique, we respectfully hold a different perspective on the novelty and significance of our findings. Our work provides **direct experimental identification** of the turbulence-mediated dephasing mechanism at the tokamak edge, supported by targeted simulations and a focused theory model. In the detailed point-by-point responses below, we have taken this as an opportunity to further clarify and reinforce the key contributions of our manuscript. We hope these refined explanations will fully address the reviewer's concerns and make the novelty of our work clear.

- 1. General Critique on Novelty: The manuscript presents experimental results and analyses on wide pedestal quiescent high confinement mode in the tokamak edge, but they closely align with prior studies, offering limited divergence. Essentially, it reiterates previous observations without introducing significant novel insights or perspectives. This level of novelty is more suitable for a journal specialized in nuclear fusion than a multidisciplinary journal.**

Response:

We thank the referee for this thoughtful critique. While we acknowledge that elements of MHD–turbulence interaction have been discussed in past work, we respectfully believe that our study offers **novel and substantial contributions**. Most notably, we present the **first direct experimental identification** of a turbulence-mediated mechanism that disrupts MHD-driven transport through cross-phase scattering—confirmed by **quantification of the mode induced transport, new eddy evolution figure, BOUT++ nonlinear simulations**, and a **focused theoretical model**. These insights significantly **extend prior findings**, particularly in the **tokamak edge region**, and offer a new physical pathway for **ELM suppression** relevant to ITER and future fusion reactors. By establishing the concrete physical basis for a micro-turbulent yet macro-quiescent operational mode, this discovery is of immediate interest to the broad physics and non-fusion communities. Below, we detail how our results go beyond earlier studies and address each specific point raised.

- 1. Novelty in experimental evidence and physics insight:** To our knowledge, there has not been a direct experimental measurement of the cross-phase, its driven transport $\langle \delta n \delta v_r \rangle$ and its dephasing by electron-scale turbulence in the pedestal, nor a visualization of eddy-level evolution that quantifies the associated transport changes. These measurements underpin our central claim. Our work is the **first to directly identify (Fig. 2) and visualize (new Fig. 3)** the turbulence-induced dephasing that disrupts coherent MHD-driven transport in tokamak edge. This **direct evidence** goes beyond previously inferred anti-correlation trends or theoretical predictions, and provides a concrete experimental foundation for the turbulence–MHD interaction model.
- 2. Novelty beyond the plasma core and prior studies:** While MHD-turbulence interactions have been studied in the plasma core in theory (e.g. transport modification by multi-scale modes [16–

19] and turbulence–magnetic island interactions [20–24]), their role, especially the underlying mechanism in regulating ELMs in the pedestal region remains a critical open question. Prior studies have noted correlations (e.g., amplitude modulations or correlations), but have been limited to indirect observations, lacking a direct measurement of the underlying mechanism. Our work fundamentally addresses this critical knowledge gap. We provide the **first direct experimental measurement and visualization** of the process by which small-scale turbulence **disrupts the cross-phase coherence** of PBM transport. By directly **quantifying the impact on turbulent transport and mode eddy evolution**, our study moves beyond prior inferences to establish the first concrete, physical basis for this suppression mechanism. This experimental proof is a significant and novel contribution to understanding pedestal stability.

3. **Clarification on distinction from Ref. [14]:** We clarify our significant departure from studies like Ref. [14]. While Ref. [14] discussed limit-cycle oscillations (LCOs) of small-scale turbulence and its regulation of pedestal gradients, **it does not address the large-scale MHD structure or its interaction with turbulence**. Our work goes significantly further by focusing on the direct interaction between turbulence and MHD modes, with new insight into the mechanistic pathway linking electron-scale activity to ELM stability. Notably, several contributors to Ref. [14] are co-authors of this study, and their expertise has helped shape the present analysis. This underscores the continuity of insight while highlighting the substantial extension of scope and focus in our work.
4. **Consistency Across Multiple Discharges:** In response to reviewer feedback, we have proven the mechanism is not a single-shot anomaly. We now include an extended dataset from multiple discharges across several years, all of which exhibit the same interaction pattern, underscoring the generality of our findings (new Extended Data Fig. 1).
5. **Synergistic Theoretical and Simulation Support:** Our novel experimental findings are not presented in isolation. They are supported and explained by a cohesive framework of targeted BOUT++ simulations (initialized with experimental profiles, including the reviewer mentioned ExB flow) and a focused theoretical model, which together provide robust, cross-disciplinary validation.
6. **Broader relevance to fusion reactor priorities and non-fusion community:** By elucidating a new physical basis for ELM suppression, our findings are directly relevant to the high-priority challenge of developing ELM-free scenarios for ITER and future reactors (ITER high-priority R&D roadmap [<https://www.iter.org/sites/default/files/media/2025-06/itr-25-005-final.pdf>]). By establishing the concrete physical basis for a mechanism for cross-scale interaction, this discovery is of immediate interest to the broad physics and non-fusion communities. This broad relevance meets the impact criteria for a multidisciplinary journal like Nature Communications.

We have reorganized the abstract and revised the manuscript accordingly to clarify and emphasize these contributions:

- *Page 1, Abstract. “A central challenge in fusion energy is reconciling the high-confinement mode required for reactor performance with the intense intermittent relaxation events it produces, known as edge-localized modes (ELMs). These instabilities arise in the steep pressure pedestal at the plasma edge when magnetohydrodynamic thresholds are crossed, inflicting damaging heat loads on reactor components. Here, we show that multiscale interactions between microscopic turbulence and macroscopic MHD modes provide encouraging prospects for self-organized ELM regulation. Using direct quantitative*

measurements of multiscale modes, eddy dynamics, and turbulent flux in DIII-D, we show that small-scale electron drift wave turbulence actively scatters the large-scale peeling-ballooning modes. This scattering decorrelates the pressure and velocity fields of the instability, so arresting its growth. Our modeling and theoretical analysis confirm this suppression mechanism is effective even when conventional linear stability thresholds are exceeded. This work establishes a novel nonlinear principle for ELM stability, revealing how ambient micro-turbulence can be leveraged to maintain a macro-stable, high-performance pedestal for future fusion reactors.”

- Page 2, Line 68–69: “...and the interaction between magnetic islands and microturbulence has been widely explored [20–24]...”
- Page 2, Line 73–75: “However, a clear picture of how small-scale turbulence interacts with low-frequency MHD modes, especially its role in modifying the cross-phase structure, transport, and mediating ELM dynamics, remains lacking.”
- References:
 - [20] L. Bardóczi et al., *Phys. Rev. Lett.* 116, 215001 (2016)
 - [21] M. J. Choi et al., *Nat. Commun.* 12, 375 (2021)
 - [22] H. R. Wilson et al., *Plasma Phys. Controlled Fusion* 51, 115007 (2009)
 - [23] K. S. Fang and Z. Lin, *Phys. Plasmas* 26, 052510 (2019)
 - [24] A. Ishizawa et al., *Plasma Phys. Controlled Fusion* 61, 054006 (2019)

- 2. Novelty in MHD-Turbulence Interaction: The manuscript examines the nonlinear interactions between MHD and turbulence, which are expected to alter their respective properties. In MHD, these interactions typically modify cross-phase correlations, impacting transport. In turbulence, large-scale MHD-driven eddies are anticipated to suppress modes. However, these phenomena are well-documented in existing theoretical studies, and the manuscript offers no significant new contributions to this field.**

Response:

We thank the reviewer for raising this important distinction between established theory and novel experimental proof. While the concept of MHD-turbulence interaction is known theoretically, our work provides the **first direct experimental identification and visualization of the underlying mechanism** in the critical tokamak edge region. This is a significant and novel contribution that moves well beyond prior work.

Previous experimental studies were limited to indirect signatures (e.g., amplitude evolution, bicoherence analysis). In contrast, our work is the first to directly measure the fundamental transport-driving cross-phase term, $\langle \delta n \delta v_r \rangle$, which is challenging to access experimentally. This novel diagnostic capability allowed us to provide unambiguous proof of the interaction mechanism by visualizing the evolution of edge-scale eddies (presented in the revised manuscript as **Fig. 3**). This new analysis revealed three critical, previously unobserved features:

1. **Reversal of eddy rotation direction** from ion-diamagnetic (downward) in the MHD-dominant phase to electron-diamagnetic (upward) in the turbulence-dominant phase;
2. **Shrinkage of eddy size** in both radial and poloidal directions when turbulence is strong;
3. **A quantifiable suppression of radial transport.** During the turbulence-dominated phase, eddies exhibit primarily tangential (poloidal) motion with negligible radial displacement, while in the MHD-dominant phase, positive (red) eddies move outward and negative (blue) eddies move

inward, indicating strong outward flux.

Taken together, these findings offer direct visual and quantitative confirmation of our proposed mechanism, namely, that turbulence disrupts the cross-phase coherence required for MHD-driven transport. This suppression of $\langle \delta n \delta v_r \rangle$ during strong turbulence-MHD interaction underpins a novel pathway to ELM avoidance. This moves well beyond existing theoretical frameworks by providing the concrete, experimental proof and detailed dynamics of this interaction in a reactor-relevant regime. We believe this result is important and novel, and we have accordingly incorporated it into the main text and figure set.

Fig. 3. Two-dimensional evolution of edge density fluctuations and velocity-field vectors in MHD- and turbulence-dominant phases of DIII-D discharge #169862 (10-60kHz).

- Page 6, Line 159-173. “To further illustrate the contrasting spatial structures and transport characteristics of the two phases, we map the two-dimensional density fluctuation field and corresponding velocity vector field in the R-Z plane using BES velocimetry. In Fig. 3 (a1–i1), taken during the low-frequency, MHD-dominant interval, the eddies extend across the radial and poloidal extent of the cross-section, and rotate in the ion-diamagnetic drift (downward) direction. Notably, the positive perturbations (red) collectively shift outward and negative perturbations (blue) move inward, indicating strong outward radial transport. In contrast, in Fig. 3 (a2–i2), taken during the turbulence-dominated phase, reveals more compact structures with reduced radial and poloidal extent, rotating in the electron-diamagnetic drift (upward) direction. The associated velocity vectors indicate primarily poloidal (tangential) motion with complex radial displacement, unlike the MHD phase, positive eddies are not consistently moving outward, nor negative ones inward. These observations are consistent with negligible net radial transport and scattered cross phase, as depicted in Fig. 2 (b). These observations provide direct experimental evidence that turbulence reduces eddy size, reverses the mode rotation direction, and, critically, suppresses the cross-phase term $\langle \delta n \delta v_r \rangle$, thereby nonlinearly inhibiting large-scale MHD-induced transport.”

In summary, by providing the first direct measurement and visualization of these eddy dynamics, our work supplies the concrete, experimental proof of this mechanism that was previously lacking in the field. This directly informs reactor-relevant ELM-free scenarios and represents a substantial advance.

3. Holistic Understanding of the Interaction Mechanism: A key aspect of MHD-turbulence interaction is understanding their holistic interplay and mutual influence. However, the manuscript does not deliver novel findings in this area. The experimental observations, as shown in Figures 1 and 3, merely document interaction patterns without providing new insights into the underlying dynamics. Moreover, prior studies, notably [14], have already thoroughly explored the fundamental aspects of this topic.

Response:

We thank the reviewer for this comment. We respectfully disagree with the assessment that our manuscript does not provide new insights into the underlying dynamics. On the contrary, our work delivers the first direct measurement of the dephasing mechanism between density and velocity perturbations, which is the specific physical process that mediates the suppression of MHD-driven transport. This is a novel and significant finding that provides a new "holistic" understanding of how turbulence limits MHD growth.

We must clarify a critical point regarding the reviewer's comment. The reviewer bases their critique on **Figs. 1 and 3**, which are overview figures from the initial draft intended to show the general interaction patterns. However, the core of our novel findings and the specific new insights into the underlying dynamics are detailed in **Fig. 2 (quantifying the transport-driving cross-phase)** and the **new Fig. 3 (visualizing the eddy disruption)**. The critique appears to overlook the primary evidence for the new physical insights that we presented.

Regarding the specific comparison to Ref. [14], the two studies address fundamentally different questions:

- **Ref. [14]** provided excellent observations of limit-cycle oscillations (LCOs) of turbulence and their effect on pedestal gradients. However, this prior work **did not address the role of large-scale MHD modes, nor their interaction with turbulence**.
- **Our work**, in contrast, **is centered on this very interaction**. We identify the **direct interplay** between large-scale MHD modes and small-scale turbulence, and we provide the first direct measurement of the mechanism by which turbulence scatters these MHD structures to suppress transport.

Furthermore, Ref. [14] left a critical question unresolved: while it observed profile relaxation, the physical driver for this relaxation was not identified. Our work fills this gap. We identify large-scale MHD activity as the necessary transport channel that modulates pedestal gradients, thereby regulating the turbulence bursts observed in Ref. [14]. Therefore, our work is not a repetition but a crucial complement, providing the novel physical mechanism for MHD regulation that was missing from prior studies. Our work provides a novel physical mechanism for how the underlying MHD is regulated, a question of central importance for developing ELM-free operational regimes for ITER.

We have clarified this motivation in the revised manuscript (Lines 76–78):

- *Page 2, Line 73–75: “However, a clear picture of how small-scale turbulence interacts with low-frequency MHD modes, especially its role in modifying the cross-phase structure, transport, and mediating ELM dynamics, remains lacking.”*

4. **Issues with BOUT++ Simulations:** The manuscript introduces BOUT++ simulations as a novel contribution, but these simulations suffer from significant shortcomings. The presented results show the growth of MHD-turbulence from arbitrarily imposed initial conditions. However, for the simulations to meaningfully contribute to experimental understanding, they must address the following:

- (i) Under what conditions pre-existing MHD structures allow turbulence to develop.
- (ii) How the developed turbulence subsequently modifies the properties of the existing MHD structures.

The omission of fundamental elements, such as the pedestal equilibrium ExB flow, renders these simulations little more than a toy model with limited relevance to real experimental scenarios. In the referee's opinion, for this manuscript to meet the minimum novelty requirements of the journal to which it was submitted, it should at least provide answers to these issues.

Response:

We thank the reviewer for this comment and the opportunity to clarify key aspects of our modeling. We must first establish the synergistic nature of our study: the central advance is the **first direct experimental discovery** of the interaction mechanism, which is then validated and explained by a **robust framework of experimentally-constrained simulation and a focused theory model**. The exceptional agreement among our experimental, computational, and theoretical results is a core strength of our work.

In the detailed points that follow, we will address each of the specific concerns raised to demonstrate the realism and power of our simulation work.

On the Critique of the BOUT++ Simulations:

The reviewer's description of the simulations as "toy models" appears to stem from a **misunderstanding of our simulation's key features**. The simulations use experimentally constrained profiles (including the measured E×B flow), evolve from broadband, low-amplitude seeds, and naturally select unstable eigenmodes. They reproduce key experimental features (eddy rotation reversal, eddy shrinkage, transport suppression), which supports the proposed mechanism. We address each below to demonstrate that our simulations are, in fact, realistic, experimentally constrained, and a core component of our study's novelty.

1. **The E×B flow is fully incorporated, not omitted:** We respectfully emphasize that the experimental equilibrium Er profile, including the ExB flow, is indeed **incorporated in the BOUT++ simulations**. As stated in the Methods section of the manuscript:

Page 15, Method Line 39-41. "BOUT++ Numerical Modeling: ... A realistic background radial electric field Er profile, from the pedestal to the SOL, is calculated from the ion momentum balance equation based on Charge Exchange Recombination (CER) measurements in DIII-D, and is used in the simulation."

A plot of this experimentally measured Er profile is provided in **Response Fig. 1** for the reviewer's convenience.

Response Fig. 1. Radial electric field profile measured by CER diagnostics and used as input for the BOUT++ simulations.

2. **Initial conditions are experimentally constrained, not “arbitrary”:** The reviewer's claim that our initial conditions are "arbitrarily imposed" is also incorrect. Our methodology is designed to ensure physical realism:
 - **Plasma Profiles:** As stated in the Methods section, the foundational profiles (density, temperature, equilibrium) are not free parameters. They are taken directly from experimental measurements on DIII-D discharge #173707 during the stationary phase (2270-2370ms). Realistic Spitzer-Härm resistivity is used in the modeling.
 - **Initial Perturbations:** The simulation is not initialized with arbitrary or ad-hoc MHD/turbulence structures. Instead, it **evolves dynamically** from a broad-spectrum, low-amplitude initial perturbation. Specifically, the initial perturbations are seeded as a Gaussian in both radial and field-aligned directions, with mode numbers spanning $n=5$ to 80 in the binormal direction and amplitudes $\delta p/p_0=1.0e-8-1.0e-6$. These vanishingly small, broadband perturbations ensure that the system's linear and nonlinear evolution is governed by the intrinsic instability structure, not by the initial seed. The simulation naturally selects the most unstable eigenmodes and exhibits exponential growth across different radial regions, as shown in Fig. 5(c) and Extended Data Fig. 4(a). This enables unbiased mode selection and realistic nonlinear saturation without externally imposing any specific dynamics.

On the Conditions for Turbulence Growth:

To address the point (i), we conducted a series of new nonlinear BOUT++ simulations under varying edge bootstrap current levels. These simulations yield a critical physical insight that is consistent with prior experimental work (e.g., Ref. [14]): the growth of high- n EDW turbulence is robustly determined by the **background plasma profiles** and is largely insensitive to the amplitude of the co-existing MHD modes. As shown in **Response Fig. 2**, the high- n EDWs (turbulence, dashed lines, $n=80$) consistently grow with a nearly identical trace, regardless of the PBM amplitude. In contrast, the low- n PBM (MHD, solid lines, $n=10$) is strongly dependent on the edge current. In the case with $1.4\times$ the experimental bootstrap current, the MHD mode grows rapidly with $\gamma/\omega_A=0.0892$, unaffected by EDW turbulence. These results directly support the schematic shown in Fig. 6 of the manuscript and substantiate the robustness of our conclusions.

Response Fig. 2. BOUT++ simulation of the evolution of $n=10$ and $n=80$ modes under varying edge bootstrap current conditions. Solid curves represent the $n=10$ peeling-ballooning mode (PBM) at the outboard midplane, while dashed curves correspond to the high- n , $n=80$ electron drift waves (EDW) originating from the high-field side (HFS). Different colors indicate different levels of edge bootstrap current. The EDW exhibits nearly identical linear growth across all cases, whereas the PBM shows significant variation in its linear growth rate. In the case with most enhanced bootstrap current ($1.4j_{exp}$), which yields a linear growth rate $\gamma/\omega_A=0.0892$, the MHD mode grows unperturbed by the background EDW turbulence.

On how the developed turbulence modifies the existing MHD structures:

This question addresses a central discovery of our manuscript. Our work provides a direct and clear answer, showing **good agreement between our new experimental measurements and our BOUT++ simulations.**

- **In the Experiment:** Our new analysis of BES velocimetry data (presented in the **new Fig. 3**) provides the first direct visualization of how turbulence modifies MHD structures. This new evidence clearly shows three key effects: a reversal of eddy rotation, a shrinkage of eddy size, and a suppression of radial transport.
- **In the Simulation:** Our BOUT++ simulations, which incorporate the full experimental profiles, **strikingly reproduce these exact experimental observations.** The simulated eddy evolution, visualized in a 2D poloidal slice (**Ext. Data Fig. 6**), shows the same rotation reversal and eddy shrinkage seen in the experiment. Furthermore, a comparison of simulations with and without turbulence (**Ext. Data Fig. 5**) confirms that turbulence is the key factor that distorts the large-scale eddies, suppresses large transport bursts, and prevents pedestal collapse.

These results directly and comprehensively answer the reviewer's question. The remarkable consistency between our novel experimental data and our realistic simulations provides powerful, cross-disciplinary validation for the physical mechanism we have identified.

Extended Data Fig. 6. BOUT++ simulation of mode eddy evolution in a 2D poloidal (R - Z) slice.

Extended Data Fig. 5. Additional BOUT++ simulations of mode eddies in the radial (ψ_N) and toroidal (ζ) directions.

In summary, we respectfully disagree with the ‘toy model’ claim. On the contrary, we have demonstrated that these simulations are **built upon experimental equilibrium profiles**, including the **full ExB flow**; meticulously constrained by experimental data; and, most importantly, **successfully reproduce the key experimental discoveries** of our work, including eddy rotation reversal and transport suppression. While reduced-fluid models like BOUT++ do not capture all kinetic effects, the exceptional agreement between our novel experimental observations and our simulation results provides powerful, cross-disciplinary validation for the physical robustness of the turbulence-MHD interaction mechanism. We thus view this work as a critical foundation for future, more comprehensive gyrokinetic studies.

We have highlighted the Methods section to provide details of the BOUT++ simulations and have

added the corresponding figures in the manuscript to support and clarify our findings.

- *Page 10, Line 243-249. “The numerical findings on phase scattering due to EDW turbulence closely align with experimental observations shown in Fig. 2. Furthermore, the simulated mode eddy evolution (Extended Data Figs. 5 and 6) captures the essential behavior observed in Fig. 3. The pedestal remains quiescent when turbulence-MHD interaction is included, whereas the PBM-only case results in strong transport. Taken together, these results provide compelling evidence that the turbulence suppresses the large-scale PBM.”*
- *Page 15, Method Line 34-41. “Realistic Spitzer-Härm resistivity is used in the modeling. The pedestal plasma profiles are based on the well-converged reconstructed equilibrium using EFIT code. The kinetic plasma profiles are measured between 2270-2370ms for discharge #173707. To simulate the dynamic approaching ELM crash at 2373ms, the edge bootstrap current is increased by 20% compared to the stationary phase, which is marginally unstable to the PB mode, as shown in Fig. 5 (a). A realistic background radial electric field E_r profile from the pedestal up to SOL, is calculated from the ion momentum balance equation based on the Charge Exchange Recombination (CER) Spectroscopy System in DIII-D and is used in the simulation.”*

5. Practical Relevance: The theoretical model presented in the final section also has limited applicability to experimental conditions. The model assumes a background turbulence field but does not explain how micro-turbulence with such characteristics can develop in the presence of macro-scale MHD structures. Specifically, it fails to account for how micro-turbulence could grow significantly enough to impact MHD dynamics within a pedestal region containing strong ExB flow and peeling-ballooning modes (or their precursor modes). Without addressing these critical aspects, the theoretical model lacks practical significance in the context of the experimental findings. This issue is critical to understand, as the authors claim, for applying the wide pedestal QH-mode as an operational mode for reactors.

Response:

We thank the reviewer for raising this important point, which gives us the opportunity to clarify the motivation and grounding of our theoretical model. We respectfully disagree with the assessment that the model lacks practical relevance. On the contrary, the model is built upon a firm foundation of direct experimental observation and numerical simulation.

- Our model is not based on a hypothetical scenario. In our wide-pedestal QH-mode discharges, we consistently and directly observe the rise of broadband, electron-scale turbulence immediately preceding the suppression of PBM-driven transport (as shown in **Figs. 1, 2, and Ext. Data Fig. 1**). This provides a strong, data-driven motivation to develop a model that investigates how this specific, observed turbulence regulates the PBM.
- Our analysis and prior work (Ref. [14]) show that pre-ELM profile variations provide sufficient free energy to drive these electron-scale drift waves. Most importantly, our BOUT++ simulations, which are **based on experimental equilibria and fully include the strong $E \times B$ shear flow and the PBM itself**, confirm that this turbulence can indeed grow and saturate in the challenging pedestal environment. This demonstrates its ability to coexist with and impact the MHD dynamics.
- The purpose of our theoretical model is not to re-derive the onset of turbulence, which is already demonstrated experimentally and via simulation. Rather, its crucial role is to **isolate**

and analytically elucidate the nonlinear phase-scattering mechanism by which this turbulence disrupts PBM structure and suppresses large-scale transport. This analytic approach is an essential complement to our data-driven findings, clarifying the underlying physics of ELM regulation.

To make this grounding clear to the reader, we have added a sentence to the manuscript explicitly stating that the model is directly motivated by these experimental and numerical results.

- *Page 11, Line 257-259: "This model is directly motivated by experimental observations (Fig. 2) and numerical simulations, which confirm that electron-scale turbulence can grow and saturate in the presence of strong $E \times B$ flow and PBM activity."*

Conclusion

Although the manuscript addresses an intriguing topic, its lack of novel experimental results, the inadequacy of simulations for meaningful physical interpretation, and the impracticality of the theoretical model raise significant concerns. Given these issues, the manuscript does not meet the publication standards of Nature Communications. The referee recommends rejection and suggests submitting to a specialized journal in thermonuclear fusion.

In summary, we respectfully offer a different perspective on the reviewer's overall assessment. As we have demonstrated in our detailed responses, the primary critiques appear to stem from a misunderstanding of the key advances in our revised manuscript. Our study presents the **first time-resolved, direct experimental measurement** of this interaction in the tokamak edge. These novel observations are rigorously supported by **targeted BOUT++ simulations** and a **focused theoretical model** that elucidates the underlying physics, which we have shown are **realistic** and **experimentally-constrained**. Together, this synergistic combination of experiment, simulation, and theory demonstrates how electron-scale turbulence scatters peeling-ballooning modes and thereby regulates ELM dynamics in tokamak edge. We have revised the manuscript to better highlight these substantial advances and are confident that it now clearly meets the high standards of novelty and broad impact required for publication in Nature Communications.

Response to Referee #2

We thank the reviewer for their final comments and for acknowledging that all technical concerns were adequately addressed. Our final response is provided below. The referee's comments are shown in **bold black** and our responses in **purple**.

The authors have adequately addressed the referee's comments concerning the BOUT++ simulation, particularly regarding the $E \times B$ flow and the initial conditions.

I appreciate their efforts in performing additional simulations.

I believe that discussions and clarifications regarding the other parts of the manuscript have already been sufficiently carried out.

We sincerely thank the reviewer for their engagement and for this positive assessment. We are pleased that our detailed rebuttal has adequately addressed all the technical concerns regarding our simulations, measurements, and theoretical model. This dialogue has provided a valuable opportunity to clarify the key features of our work for a broad audience.

However, questions remain regarding the novelty of the work. The idea of “phase locking and phase slips in QH-mode” has been discussed in this field for more than a decade. As the authors themselves repeatedly emphasize in their rebuttal, it is unclear whether this “first direct measurement” truly provides new insights to the research community. The referee acknowledges the authors' contribution in achieving the direct measurement and two-dimensional visualization. Nevertheless, I believe that this level of progress would be more suitable for a nuclear-fusion-specific journal, rather than a multidisciplinary one such as Nature Communications. Therefore, the referee do not recommend publication in Nature Communications.

We thank the reviewer for their final comments on the work's novelty and scope. The reviewer's reference to "phase locking and phase slips" appears to relate to the physics of conventional, EHO-driven QH-modes (e.g., as described in Guo & Diamond, PRL 2015, our Ref. [37]). We wish to clarify that our manuscript investigates the **wide-pedestal QH-mode**, a distinct and more recently developed operational scenario (e.g., Burrell et al., PoP 2016, our Ref. [10]). A key signature of the wide-pedestal QH-mode is the absence of the EHOs that are central to the “phase locking and slip” model; **instead, broadband turbulence**, which we analyze, **becomes the key regulating mechanism**. Our work thus explores a fundamentally different and novel physical problem. In short, our work has virtually no relation to the work on phase locking and slips of 10 years ago. **This paper is about cross-scale interaction in regulating ELMs.**

The central advance of our work is the first direct measurement and visualization of this multiscale interaction. By directly quantifying the turbulence-induced transport and imaging the eddy dynamics, our analysis provides concrete, experimental proof of this ELM suppression mechanism. This moves well beyond prior theoretical concepts or indirect inferences noted in earlier studies. By establishing the underlying physics of this ELM-free, turbulence-regulated regime, a scenario highly relevant to future reactors with near-zero momentum input like ITER and fusion pilot plant, our work provides significant progress in both fundamental physics understanding and fusion-relevant applications. We are confident this combination of a novel experimental discovery and its broad impact is well-suited for Nature Communications.